# SLGA-YOLO: A Lightweight Castings Surface Defect Detection Method Based on Fusion-Enhanced Attention Mechanism and Self-Architecture

**DOI:** 10.3390/s24134088

**Published:** 2024-06-24

**Authors:** Chengjun Wang, Yifan Wang

**Affiliations:** School of Artificial Intelligence, Anhui University of Science and Technology, Huainan 232001, China; 2001055@aust.edu.cn

**Keywords:** castings’ surface-defect detection, YOLOv8, attention mechanism

## Abstract

Castings’ surface-defect detection is a crucial machine vision-based automation technology. This paper proposes a fusion-enhanced attention mechanism and efficient self-architecture lightweight YOLO (SLGA-YOLO) to overcome the existing target detection algorithms’ poor computational efficiency and low defect-detection accuracy. We used the SlimNeck module to improve the neck module and reduce redundant information interference. The integration of simplified attention module (SimAM) and Large Separable Kernel Attention (LSKA) fusion strengthens the attention mechanism, improving the detection performance, while significantly reducing computational complexity and memory usage. To enhance the generalization ability of the model’s feature extraction, we replaced part of the basic convolutional blocks with the self-designed GhostConvML (GCML) module, based on the addition of p2 detection. We also constructed the Alpha-*EIoU* loss function to accelerate model convergence. The experimental results demonstrate that the enhanced algorithm increases the average detection accuracy (mAP@0.5) by 3% and the average detection accuracy (mAP@0.5:0.95) by 1.6% in the castings’ surface defects dataset.

## 1. Introduction

Castings are metal parts created using various casting methods. Traditional casting involves injecting liquid metal into a model and solidifying the parts after cooling, also known as the liquid forming method. Currently, to meet precision requirements, specialized casting technology is commonly utilized, such as metal mold casting, pressure casting, vacuum suction casting, and other rapid forming methods [1]. Due to the complexity and diversity of the casting process, surface defects of castings are unavoidable. Castings’ surface defects can take various forms, such as cracks, blowholes, porosities, and sand holes. These defects can have severe consequences for subsequent equipment processing, shortening the service life and posing risks to both the machine and users. Therefore, Sur-face Defect Detection on Castings has been one of the research focuses [2]. 

Traditional methods for defect detection rely on human visual checking, which utilizes machine vision devices, such as industrial cameras and lenses, for identification and detection. These devices use light sources and image-acquisition equipment to collect images and determine whether there are defects. The process involves five steps: image preprocessing, feature extraction, template matching and positioning, and positive and negative classification [3]. Currently, intelligent equipment manufacturing companies typically use Cognex’s machine vision software, VisionPro(9.0), due to its flexibility, power, and a collection of many image-processing algorithms [4]. However, traditional methods of defect detection are mainly based on manually designed features and conventional machine learning. This approach has limitations in complex scenes compared to deep learning. Furthermore, it heavily relies on manual work, which can be influenced by the inspector’s experience, technical skills, and proficiency, resulting in low efficiency and accuracy. Additionally, long working hours may cause visual fatigue among inspectors, leading to missed detections and false alarms.

In recent years, with the continuous development of deep learning, deep learning-based defect detection methods have gradually replaced traditional detection methods. Currently, there are two main types of deep learning-based detection algorithms: one-stage algorithms, such as SSD [5] and YOLO [6,7]; and two-stage algorithms, such as Mask R-CNN [8]. Two-stage algorithms have been used for detecting long-distance small targets [9], and one-stage algorithms have been used for fast identification and detection [10], as well as wire insulator fault detection and foreign-object detection [11]. Lan et al. [12] proposed a new lightweight model named Swin-T YOLOX, which consists of the advanced detection network YOLOX [13] and a robust backbone Swin Transformer [14]. Hurtik et al. [15] presented a new version of YOLO called Poly-YOLO, which builds on YOLOv3 and addresses issues by aggregating features from a lightweight SE-Darknet-53 backbone using a hypercolumn technique, employing stairstep upsampling, and generating a single-scale output with high resolution. Compared to YOLOv3, Poly-YOLO has only 60% of its trainable parameters but improves the mean average precision by 40%.

In conclusion, the casting surface-defect detection technology has been developed for a long time, and the traditional detection technology methods have been gradually replaced by the current deep learning-based methods. Although researchers have conducted a lot of research on the key issues, such as high accuracy and lightweight, the determination of how to achieve high precision and lightweight coexistence still faces challenges.

This study proposed a castings surface defect detection method based on the YOLOv8 algorithm, named SLGA-YOLO, which improved on the basic model of YOLOv8. Firstly, the neck module is optimized using SlimNeck, which can significantly reduce the number of parameters while maintaining sufficient accuracy. Secondly, SimAM and LSKA are fused in the backbone to strengthen the attention mechanism and improve the algorithm’s focus on the focal region. Then, to achieve high accuracy while maintaining the requirement of meeting the lightweight requirement, YOLO-P2 is used to replace the original model to achieve four-detector-head coexistence detection, based on which a novel GCML module is designed to replace a part of the base convolutional block, CBS to make full use of the extracted feature information. Finally, the Alpha-*EIoU* loss function is constructed to maintain sufficient flexibility and strong generalization. The main improvements are summarized as follows:
SlimNeck is used to optimize the model neck module, reducing model complexity whilst increasing accuracy.The fusion of SimAM and LSKA strengthens the attention mechanism to enhance the three-bit weight extraction capability of the model, as well as to enhance the multiscale feature extraction capability.YOLO-P2 is used to replace the original model to improve the efficiency of model detection of small targets. At the same time, some of the basic convolutional blocks (CBSs) are optimized as self-defined design GCML modules to improve the convergence speed of the model and enhance the feature extraction capability.The Alpha-*EIoU* loss function is constructed to accelerate the regression fitting process of the real frame and prediction frame, thus maintaining sufficient flexibility and strong generalization.


## 2. Related Works

### 2.1. Traditional Defect Detection

Defect detection methods based on traditional machine vision are typically divided into several steps, including image preprocessing, threshold segmentation, and feature extraction. The processed feature image is then used to determine whether there are any defects in the current product.

Yuan et al. [16] proposed an improved Otsu threshold segmentation method called Weighted Object Variance (WOV), which obtains 94% DR and 8.4% FAR of the defect detection for tested rail images. Capizzi et al. [17] proposed a GLCM-based detector for fruit defects using a radial basis probabilistic neural network; this model can reduce the overall error of detection to 2.75%. Li et al. [18] proposed an algorithm for detecting surface defects in steel bars based on LAC. The algorithm achieves a detection success rate of over 95%, and the false-alarm rate is lower than 7%. In the defect inspection of castings, Li et al. [19] proposed two methods for processing images of defects in castings using traditional X-ray inspection. While the methods can automatically detect internal defects in castings, they require manual operation in practical applications and cannot detect defects with non-obvious features.

### 2.2. Deep Learning-Based Defect Detection

As the industrial scale grows, higher demands are placed on algorithm performance. Deep learning is an emerging field that has evolved to address the need for generalization and hyper-complexity in problem-solving without a significant increase in computational cost.

Tang et al. [20] proposed a non-destructive defect detection for castings based on a self-developed convolutional neural network, called spatial attention bilinear convolutional neural network, which achieved an accuracy of more than 90% on the Xdefects data set. Hao et al. [21] proposed a method for detecting casting defects using an improved DETR. The method establishes a detection model through multi-scale feature fusion and an improved backbone network. The results show that the model achieves an accuracy of 49.6%. Jiang et al. [22] proposed a casting appearance detection method based on the deep residual network, selecting ResNet-34 as the network model and improving the activation function. The experimental results show an improvement of 8.8% compared to the traditional method. Du et al. [23] proposed a method for improving X-ray defect detection using deep learning. The method is based on Faster R-CNN and has been enhanced at the algorithmic level. Experimental results demonstrate that the detection effect is superior to the baseline Faster R-CNN. Parlak et al. [24] proposed a method for detecting defects and their types in aluminum castings using deep learning. He analyzed the accuracy of different algorithms and found that YOLOv5 achieved a detection accuracy of 95.9%.

### 2.3. Summary

The preceding paragraphs summarize the research related to traditional defect detection and deep learning-based defect detection. It is demonstrated that, although the existing deep learning-based algorithmic detection has significantly improved the detection efficiency and accuracy, it is unable to meet the target detection in all fields, especially for castings’ surface-defect detection, which is not specialized enough to meet the needs of the castings’ manufacturing producers. Furthermore, the existing algorithmic models are unable to achieve lightweight while ensuring accuracy for deployment in low-performance equipment. To address these issues, this paper proposes a novel lightweight inspection model, SLGA-YOLO, designed for the specific purpose of detecting casting surface defects. 

## 3. Design for SLGA-YOLO

In order to meet the requirements of fast speed and high precision for casting surface-defect detection, the improved algorithm model (SLGA-YOLO), as shown in Figure 1, included three major parts: the backbone, the neck, and the predicted head.

Firstly, by employing the SlimNeck constructed based on GSConv and VoV-GSCSP to replace the neck part of the model, i.e., connecting the standard backbone to the SlimNeck, we aimed to enhance the model’s running speed and reduce the computational complexity, enabling it to detect defects promptly. Secondly, due to the mixed defects in the collected casting surface-defect dataset, the ability to extract key feature information is weak. To solve this problem, we integrated SimAM and LSKA to strengthen the attention mechanism: the SimAM was incorporated into the backbone network to make the network focus on key information during training. Embedding the LSKA in the SPPF improved the model’s perceptual range of input features. In addition, based on the addition of the p2 detection layer, we propose a lightweight model called GCML to replace part of the base convolutional block (CBS) in order to reduce the redundancy of the feature maps present in the neural network and better exploit the feature information. Finally, to accelerate model convergence while providing enough flexibility to improve model accuracy, we designed the Alpha-*EIoU* loss function. In comparison to YOLOv8, this method provides an effective balance between model lightweight and model accuracy.

### 3.1. Improved YOLOv8 Neck Module Based on SlimNeck

In actual defect detection, the algorithm detection speed is required. The neck is situated between the backbone and the predicted head and is employed primarily for feature fusion and enhancement. In contrast, SlimNeck is capable of achieving an optimal balance between feature fusion and enhancement, accuracy, and speed. To improve the detection speed without reducing the accuracy, an improved method based on SlimNeck [25] is proposed. 

The YOLOv8 model utilizes the CBS standard convolutional block and C2F module in the neck as the feature information processing module. However, this module fails to balance the accuracy and speed of the model. To improve the accuracy and reduce the number of model parameters without reducing the feature fusion capability, the GSConv convolution block and VoV-GSCSP module from SlimNeck were introduced to replace the CBS and C2F in the original model. GSConv is a lightweight convolution that is computationally inexpensive and has a high feature fusion capability compared to CBS.

The input undergoes standard convolution and deep convolution. The results of both convolutions are then combined through splicing, and a shuffle operation is performed to match the corresponding channels. Finally, the output is produced. VoV-GSCSP utilizes a cross-layer network structure with fewer parameters, maintaining accuracy while reducing computational complexity and network structure. The structures of VoV-GSCSP, GSConv, and DWConv can be seen in Figure 2.

In conclusion, we propose using GSConv-based SlimNeck to replace the neck part of the model. Using the GSConv convolution operation is one of the main ways to reduce redundancy and duplicate information without the need for compression, resulting in reduced inference time while maintaining accuracy. By replacing the neck part with VoV-GSCSP, we aim to convey deep semantic features from the top down, reduce the number of modular parameters, and reduce the computational burden.

### 3.2. Attention Module

The basic YOLOv8 model algorithm belongs to the one-stage detection algorithm, with improved detection speed, and is linked by multiple CBS and C2F structures in the backbone network to increase the model’s ability to extract graphical feature information. However, since the model’s ability to extract input features is weak, it is difficult to dig deep into details, which means that it often tends to ignore deep feature information, thus reducing the model’s sensitivity to small targets, and the extraction of effective feature information is incomplete. At the same time, there is a large amount of redundant information extraction, which increases the computational burden; it is unable to effectively capture the global relationship of the data, and it is weak in the ability to fight against noise interference, of which, the backbone network is the basis of the model and is responsible for extracting features from the input image. These features are the prerequisite for the subsequent network layers for target detection. To solve such problems, we propose a novel fusion attention enhancement mechanism that uses the joint action of LSKA and SimAM to further improve performance, reduce the role of interfering information, continuously focus on the key region information, and improve the detection ability of the algorithm. By introducing the fusion of LSKA and SimAM to enhance the attention mechanism, SLGA-YOLO can pay more attention to the key features in the defect images.

#### 3.2.1. LSKA Attentional Mechanism

LSKA (Large Separable Kernel Attention) [26] is an improvement of the LKA (Large Kernel Attention) [27] module. The LKA module has been shown to provide good performance in multi-class target detection tasks. However, the depth-wise convolution layer in the LKA module leads to computational inflation. To address this issue, LSKA suggests breaking down the 2D convolution kernels of the depth-wise convolution layer into a series of connected 1D convolution kernels. These can then be used in the attention module with a large convolution kernel of the depth-wise convolution layer. This approach can significantly reduce the amount of computation required without impacting performance. The experimental results show that the proposed LSKA attention mechanism significantly reduces computational complexity and memory footprint with increasing kernel size and provides improved performance in areas such as detection and recognition compared to the LKA module. The structure is illustrated in Figure 3a.

The output of LSKA attention mechanism is formulated as shown in Equations (1)–(4).
(1)$${\bar{Z}}^{C}=\underset{H,W}{\sum }W_{(2d-1)\times 1}^{C}*(\underset{H,W}{\sum }W_{1\times (2d-1)}^{C}*F^{C})$$
(2)$$Z^{C}=\underset{H,W}{\sum }W_{\lfloor \frac{k}{a}\rfloor \times 1}^{C}*(\underset{H,W}{\sum }W_{1\times \lfloor \frac{k}{a}\rfloor }^{C}*{\bar{Z}}^{C})$$
(3)$$A^{C}=W_{1\times 1}*Z^{C}$$
(4)$${\bar{F}}^{C}=A^{C}⊗F^{C}$$
where $F^{C}$ is a given input feature map; *H* and *W* are the height and width of the feature map, respectively; and *C* is the number of input channels. Moreover, * and ⊗ represent convolution and Hadamard product, respectively; $Z^{C}$ is the output of deep convolution;$A^{C}$ is the attention map; d is the expansion rate; and k is the maximum sensory field.

SPPF is an improvement based on SPP [28], which is a simpler and less computationally intensive model than the original structure. The SPPF module is used in YOLOv8 to capture multi-scale information by performing various degrees of pooling operations on the input feature maps. The LSKA attention mechanism enables the model to focus on important parts of the input features, and it improves the overall performance and efficiency. To enhance the model’s performance and capture the global feature relationships within the feature map, we introduce the attention mechanism LSKA after the Concat so that the LSKA attention mechanism can be utilized after SPPF completes operation. This improves the model’s perceptual range and ability to model input features. The structure is shown in Figure 3b.

#### 3.2.2. SimAM Attention Mechanism

SimAM [29], or Simple Attention Mechanism, is a lightweight and parameter-less attention mechanism based on neuroscience theory. It is designed with an energy function to achieve its purpose, as shown in Equation (5).
(5)X~=sigmoid(1E)⊙X
where E is the energy function; X is the input feature layer.

The embedded attention mechanism in the model does not introduce any additional parameters. It can directly estimate the weights of the three-dimensional features, resulting in a faster inference speed compared to attention modules such as CBAM [30]. Additionally, it improves the baseline model performance with stable results; the visual representation of this mechanism is shown in Figure 4.

The features of casting surface defects are not easily discernible, making it challenging to extract crucial information. The SimAM can adaptively adjust the feature mapping weights and pay more attention to the local area, which can improve the feature representation and classification ability of the model, while the neck will perform a Concat operation at layers 4, 6, and 9 of the backbone. Therefore, the introduction of SimAM afterward can make the model improve the feature extraction and fusion ability without increasing the computational burden again. By introducing SimAM, SLGA-YOLO can pay more attention to key features related to casting surface defects in the image, better capture fine-grained details, and thus improve detection accuracy. This enhancement enables the model to better handle the difficulties encountered in the casting surface-defect detection task, such as changes in environmental conditions, lighting, and viewing angles.

### 3.3. Optimization Model Based on GCM Module

To enhance detection accuracy, we incorporated the YOLOv8-p2 detection layer. However, this led to an increase in computational complexity due to the additional detection head. To balance accuracy and lightweight, we developed our own GCML module based on Ghostnet. The GhostConv module first aggregates the information features between channels and then employs grouped convolution to generate a new feature map. This approach allows for the generation of more feature maps with less computation. However, to ensure that the generated feature maps retain key feature information, we incorporate the LSKA attention mechanism before the splicing operation. This mechanism enables the module to focus on the key information in the input feature maps, while ignoring most of the influencing factors. As shown in Figure 5, part of the CBS module is replaced by the lightweight GCML (GhostConv-Mish-LSKA) module. It consists of the GC [31] (GhostConv) module, which is based on an improved CBM convolutional block, and the LSKA attention module.

Activation functions are a crucial component of neural networks in deep learning. They are responsible for the nonlinear transformation of a neuron’s output, which enhances the network’s expressive and nonlinear modeling abilities. YOLOv8 utilizes SiLU as its activation function. While SiLU is smoother than ReLU, it only exhibits advantages in multi-layer neural networks due to the limitations of the Sigmoid function. Due to its soft saturation, problems such as gradient vanishing may arise, leading to slow learning. To address these issues, we implemented the Mish [32] function as the activation function for SLGA-YOLO. Mish has several advantages that help to solve problems such as gradient vanishing, strong generalization, and smoother transitions. The expression of Mish can be found in Equations (6) and (7).
(6)Mish(x)=xtanh(softplus(x))
(7)$$softplus(x)=\ln(1+e^{x})$$

In the algorithm for detecting surface defects in castings, the feature input passes through the BN layer and is then processed by the Mish function. The improved convolution block CBM is shown in Figure 6.

### 3.4. Build Loss Function Based on Alpha-EIoU

The loss function is the discrepancy between the predicted and actual values following model analysis. The bounding box regression loss function is crucial for accurate detection results. In YOLOv8, *IoU* is used as a standard to measure target detection accuracy. The function can be represented by Equation (8).
(8)$$IoU=\frac{\left|A\cap Β\right|}{\left|A\cup Β\right|}$$
where *A* represents the area of the prediction box, *B* represents the area of the real box, and *IoU* represents the ratio of the intersection area to the common area between the prediction box and the real box.

The surface defects of castings are complex and varied, and the defect area is different. Therefore, the detection algorithm needs better positioning results and faster convergence speed. The *IoU* loss function cannot effectively describe the target of bounding box regression, resulting in inaccurate results and slow convergence, *EIoU* [33]. The aspect-ratio influence factors of the predicted box and the real box were separated, and the length and width of the predicted box and the real box were calculated, respectively, which met the requirements. Alpha [34] is a power parameter that provides flexibility for different levels of regression accuracy. To make the loss function more robust and flexible, Alpha-*EIoU* as the model bounding box regression loss function is reconstructed. The Alpha-*EIoU* loss function can expedite the regression fitting process of the real frame and prediction frame, thereby maintaining ample flexibility and robust generalization. *EIoU* and Alpha-*EIoU* are described in Equations (9) and (10).
(9)$$\begin{array}{ll} l_{EIoU} & =l_{IoU}+l_{dis}+l_{asp}\\ & =1-IoU+\frac{\rho^{2}(b,b^{\mathrm{gt}})}{c^{2}}+\frac{\rho^{2}(w,w^{gt})}{C_{w}^{2}}+\frac{\rho^{2}(h,h^{gt})}{C_{h}^{2}} \end{array}$$
(10)$$\begin{array}{ll} l_{\alpha -EIoU} & =l_{\alpha -IoU}+l_{\alpha -dis}+l_{\alpha -asp}\\ & =1-{IoU}^{\alpha }+\frac{\rho^{2\alpha }(b,b^{\mathrm{gt}})}{c^{2\alpha }}+\frac{\rho^{2\alpha }(w,w^{gt})}{C_{w}^{2\alpha }}+\frac{\rho^{2\alpha }(h,h^{gt})}{C_{h}^{2\alpha }} \end{array}$$


There are three components in the formula: *IoU* loss, distance loss, and height-width loss; $C_{w}$ and $C_{h}$ are the width and height of the minimum bounding box between the prediction box and the real box, ρ is the Euclidean distance between two points, b and $b^{\mathrm{gt}}$ are the centroids of the prediction box and the real box, *w* and *h* denote the width and height of the prediction box, $w^{gt}$ and $h^{gt}$ denote the width and height of the real box, and α denotes the parameter size.

## 4. Experimental Results and Analysis

In this section, we perform data augmentation on the original datasets to enhance its robustness. Furthermore, we evaluate the improved model and the classical model on the enhanced datasets to demonstrate the superiority of the former.

### 4.1. Dataset Processing

Due to the limited availability of publicly accessible casting-defect datasets both domestically and internationally, we chose to use a self-constructed dataset for our experiment. This enabled us to verify the effectiveness of our improved detection algorithm on the datasets. The casting surface-defect data were collected using an industrial camera, capturing a total of 2665 images from various angles, castings positions, and backgrounds. The open-source online annotation tool, Make Sense, is used in the data-processing process to annotate images of castings’ surface defects. The defect labels can be found in Table 1.

To solve the small number of original datasets, the robustness is low. After the image annotation is completed, the original dataset is expanded using data enhancement methods, and the final dataset totals 4020 sheets. The format of the dataset is the COCO data format; the annotation file is of .xml type; and the annotated data are finally randomly divided into training set, testing set, and validation set in the ratio of 8:1:1. The labels of the enhanced datasets are shown in Table 2.

### 4.2. Training Environment Setup

The experimental environment was configured as follows during the experiment.

An Intel(R) i5-12490F Processor and an NVIDIA GeForce RTX3060-12g were used for this study. The deep learning model was built using PyTorch 2.0.0 and Python 3.8, with a CUDA version of 11.7, and the operating system was Windows 10. To ensure fairness and comparability of model effects, pretraining weights were not used. During the training of the network model, the Mosic and Mix-up data enhancement scale was set to (1.0, 0.0), and the batch size was set to 16. Further details regarding the model’s training phase can be found in Table 3.

### 4.3. Assessment of Indicators

The experiments were conducted using commonly used evaluation metrics for defect detection to evaluate the model, including *P* (Precision), *R* (Recall), and mAP (Mean Average Precision).
(11)$$P=T_{P}\cdot {\left(T_{P}+F_{P}\right)}^{-1}$$
(12)$$R=T_{P}\cdot {\left(T_{P}+F_{N}\right)}^{-1}$$
(13)$$mAP=\frac{1}{n}\sum_{i=1}^{n}\int_{0}^{1}P_{i}(R_{i})dR_{i}$$
where $T_{P}$ represents true positive, $F_{P}$ represents false positive, $F_{N}$ represents false negative, and n represents the total number of classes. The mAP value is a measure of the accuracy of the prediction model, with higher values indicating greater accuracy. 

This study evaluated the mAP values for each category at the intersection and union threshold of mAP@0.5 and mAP@0.5:0.95 for both real and predicted frames. The mAP values for each category were evaluated at the intersection and concurrence threshold of real and predicted frames, ranging from 0.5 to 0.95. Additionally, the number of parameters, FLOPs, and FPS were also included as evaluation metrics.

### 4.4. Comparison of Experimental Results

#### 4.4.1. Ablation Experiments

To evaluate the performance of the model, we evaluated the improvements through ablation experiments to determine the feasibility of the improved model. Table 4 and Figure 7 provide the results of the experiments on our dataset.

A: YOLOv8n (baseline). B: YOLOv8-p2, represents the YOLOv8 model with the p2 detection layer introduced. Although the average detection accuracy has slightly improved, the amount of floating-point computation has significantly increased. C: YOLOv8-p2-SimAM, represents the addition of the SimAM attention mechanism on top of YOLOv8-p2, which improves detection accuracy without increasing computational burden. D: YOLOv8-p2-SimAM-GCML, Model D replaces the CBS module with the GCML module and integrates it with the LSKA attention mechanism. E: SLGA-YOLO, created by adding the SlimNeck optimized neck module to previous models and constructing the Alpha-*EIoU* loss function. The results demonstrate that SLGA-YOLO significantly reduces the number of floating-point operations and parameters while improving the average detection accuracy. The detection speed also meets practical requirements.

#### 4.4.2. Experimental Results with an Improved Version of YOLOv8 (SLGA-YOLO)

The experiments were conducted using the same GPU and enhanced datasets for the YOLOv8 algorithm before and after the improvement. The results indicate that the SLGA-YOLO has an improved average detection accuracy (mAP@0.5) of 3% and an improved average detection accuracy (mAP@0.5:0.95) of 1.6%.

Figure 8 compares the mAP values and P-R curve results before and after improving the YOLOv8 algorithm. The results demonstrate an improvement in defect detection accuracy with the improved model. Figure 9 displays the detection result of various casting defects in a laboratory environment, indicating that the SLGA-YOLO enhances detection efficiency and reduces misdetection rates. In conclusion, the proposed improved YOLOv8 algorithm performs well. Figure 10 demonstrates the comparison of the effect before and after the improvement in the actual testing.

#### 4.4.3. Comparison of Experiments on Public Datasets

To assess the model’s capacity for generalization, the publicly available dataset utilized in this study is derived from the NEU surface defect database [35], which encompasses 1800 grayscale images, with a total of 300 samples for each of the six distinct surface defect types (Cr, Pa, In, PS, RS, and Sc). The dataset is divided into three subsets, training (80%), validation (10%), and testing (10%), according to the ratio of 8:1:1. The experimental environment and configuration remain unchanged, and given that our model is improved based on YOLOv8, which is optimized on YOLOv5, we used the SLGA-YOLO model, the baseline YOLOv8n model, and the YOLOv5n model for training and validation. The validation results are presented in Table 5, and the training results are shown in Figure 11.

As demonstrated in the accompanying Table 5 and Figure 11, the improved YOLOv8 model (SLGA-YOLO) yielded favorable outcomes in this study. In comparison to Yolov5n and the baseline YOLOv8n, mAP@0.5, GFLOPs, and FPS were enhanced, while the number of parameters, although marginally higher than YOLOv5n, was less than the baseline YOLOv8n. The experimental results show that our model not only exhibits high accuracy but is also advantageous in terms of computational complexity and possesses strong generalization ability. Thus, the SLGA-YOLO model is suitable for casting surface-defect detection and can be deployed in embedded devices or mobile devices.

#### 4.4.4. Comparison of Results of Different Models Experiments

To illustrate the higher accuracy of our model and to validate the superiority of our model on the self-constructed casting surface-defect dataset, we compared our model with current popular detection algorithm models (including YOLOv5n, YOLOv8n, Mask-RCNN [8], Faster-RCNN [36], TOOD [37], DINO [38], and RetinaNet [39]). All experimental environments and configurations were kept constant throughout the comparison experiments, and the comparison experiments were conducted based on the MMDetection framework (MMDetection is an open-source target detection toolkit in PyTorch). The results of the experiments are shown in Figure 12 and Table 6. As shown in Figure 12, the improved model (SLGA-YOLO) consistently outperforms the other models in terms of average detection accuracy. In addition, as shown in Table 6, our model exhibits higher accuracy, while reducing the number of parameters compared to other models.

The experimental results demonstrate that the SLGA-YOLO model proposed in this study exhibits a superior performance compared to other algorithms of a similar nature. It reduces the number of parameters and computational burden, while maintaining the accuracy improvement, and is therefore suitable for integration into the real-time castings production line.

## 5. Conclusions

This paper introduces SLGA-YOLO, a lightweight algorithm that addresses the issues of the high computational cost, large model size, and high leakage rate in casting surface-defect detection. The model incorporates the SlimNeck optimization model neck module to reduce model complexity. Additionally, the model integrates the SimAM and LSKA fusion enhancement attention mechanism to enhance attention to important information. Additionally, we propose the GCML module to enhance the model’s understanding of input features comprehensively. Furthermore, we reconstructed the novel bounding box loss function Alpha-*EIoU* to provide the model with sufficient flexibility and strong generalization. The results demonstrate that the average detection accuracy is improved to 86.2%, and there is a drastic reduction in the number of algorithm parameters. SLGA-YOLO establishes a reliable foundation for the field of surface defect detection in castings. However, SLGA-YOLO has some shortcomings, such as the need to improve detection accuracy. Future work will focus on enhancing its adaptability to different scenarios and environmental conditions. Additionally, the algorithm’s inference speed will be optimized by exploring more sophisticated deep learning techniques and incorporating the latest developments in the field.

## Figures and Tables

**Figure 1 sensors-24-04088-f001:**
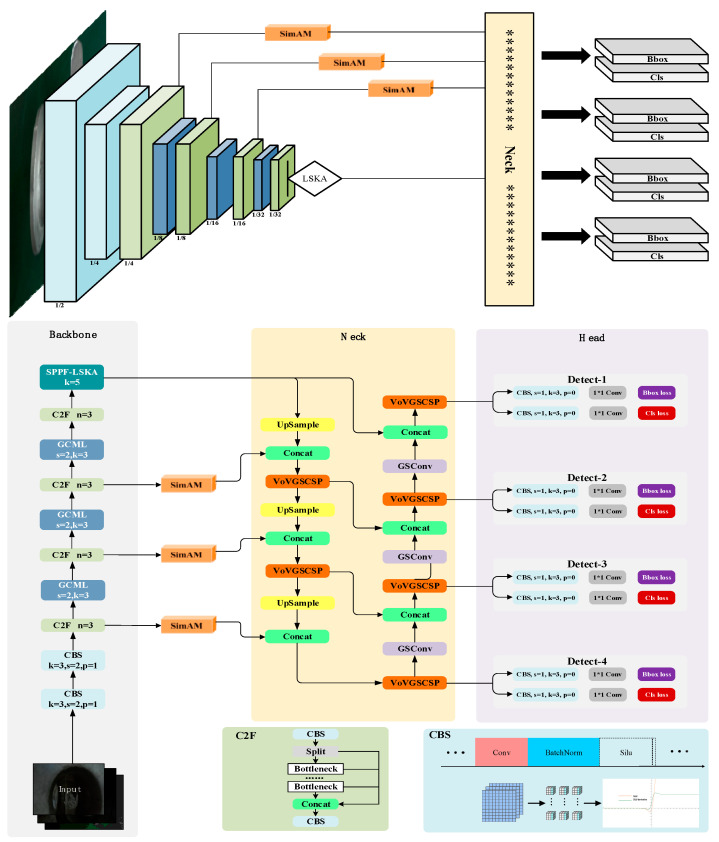
SLGA-YOLO network structure diagram.

**Figure 2 sensors-24-04088-f002:**
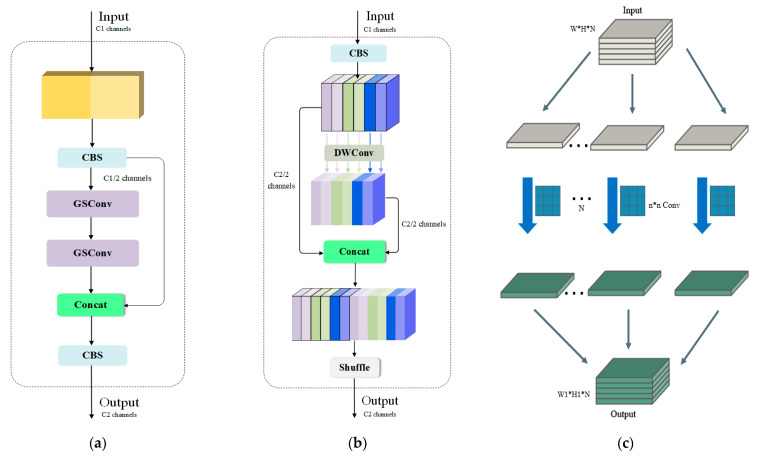
The structures of (**a**) GSConv, (**b**) VoV-GSCSP, and (**c**) DWConv.

**Figure 3 sensors-24-04088-f003:**
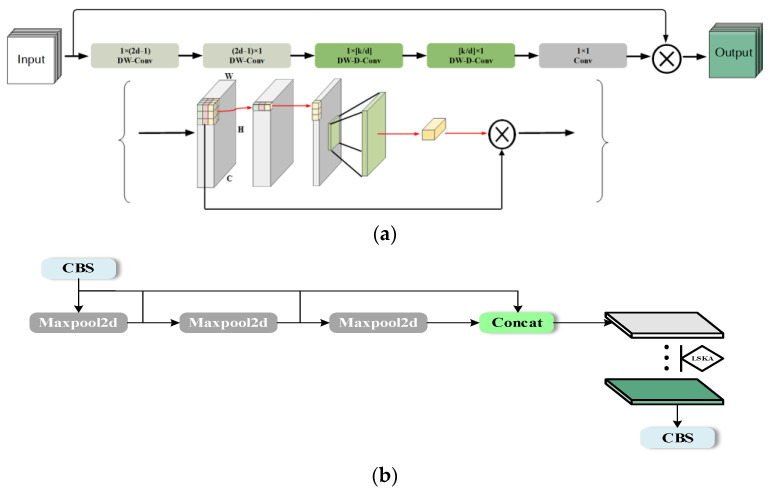
The structures of (**a**) LSKA attention mechanism; ⊗ represents Hadamard product, k represents the maximum receptive field, and d represents the dilation rate. (**b**) SPPF-LSKA.

**Figure 4 sensors-24-04088-f004:**
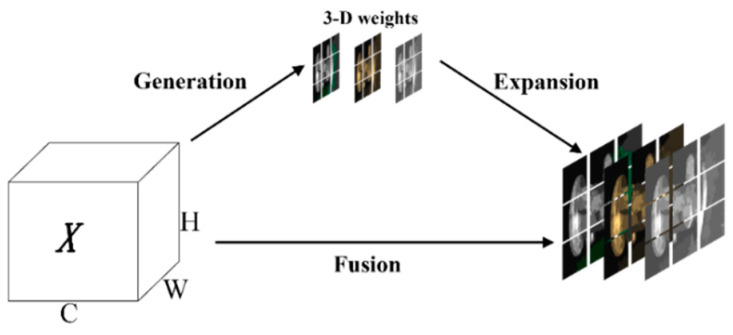
SimAM attention module structure.

**Figure 5 sensors-24-04088-f005:**
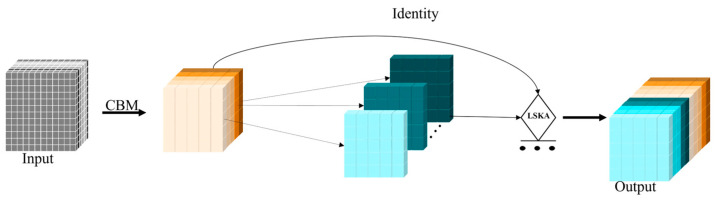
The structure of GCML.

**Figure 6 sensors-24-04088-f006:**
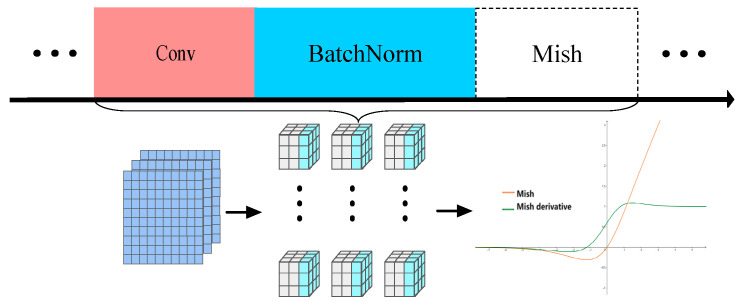
The structure of CBM.

**Figure 7 sensors-24-04088-f007:**
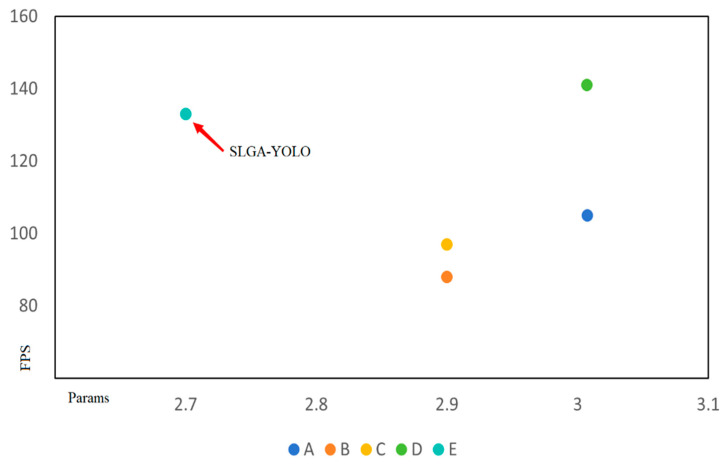
Scatter plot of comparative experimental results.

**Figure 8 sensors-24-04088-f008:**
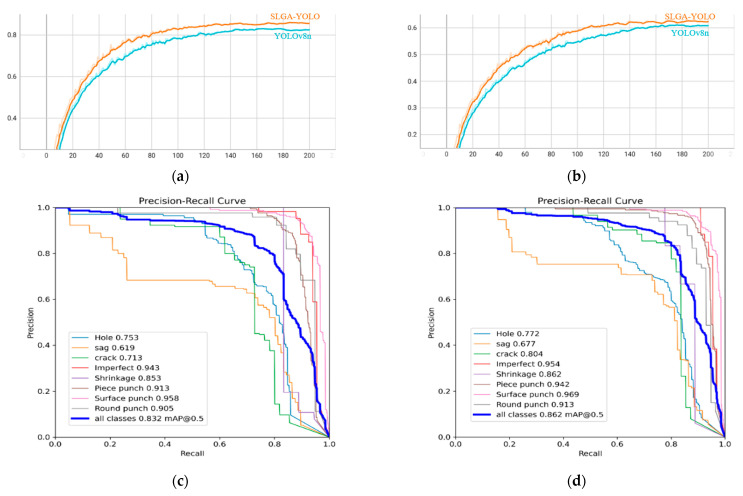
Comparison of detection results before and after improvement: (**a**) mAP@0.5, (**b**) mAP@0.5:0.95, (**c**) P-R YOLOv8, and (**d**) P-R improved YOLOv8 (SLGA-YOLO).

**Figure 9 sensors-24-04088-f009:**
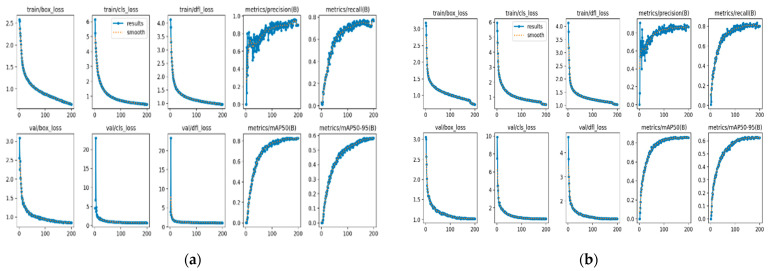
The comparison of detection results: (**a**) YOLOv8 and (**b**) improved YOLOv8 (SLGA-YOLO).

**Figure 10 sensors-24-04088-f010:**
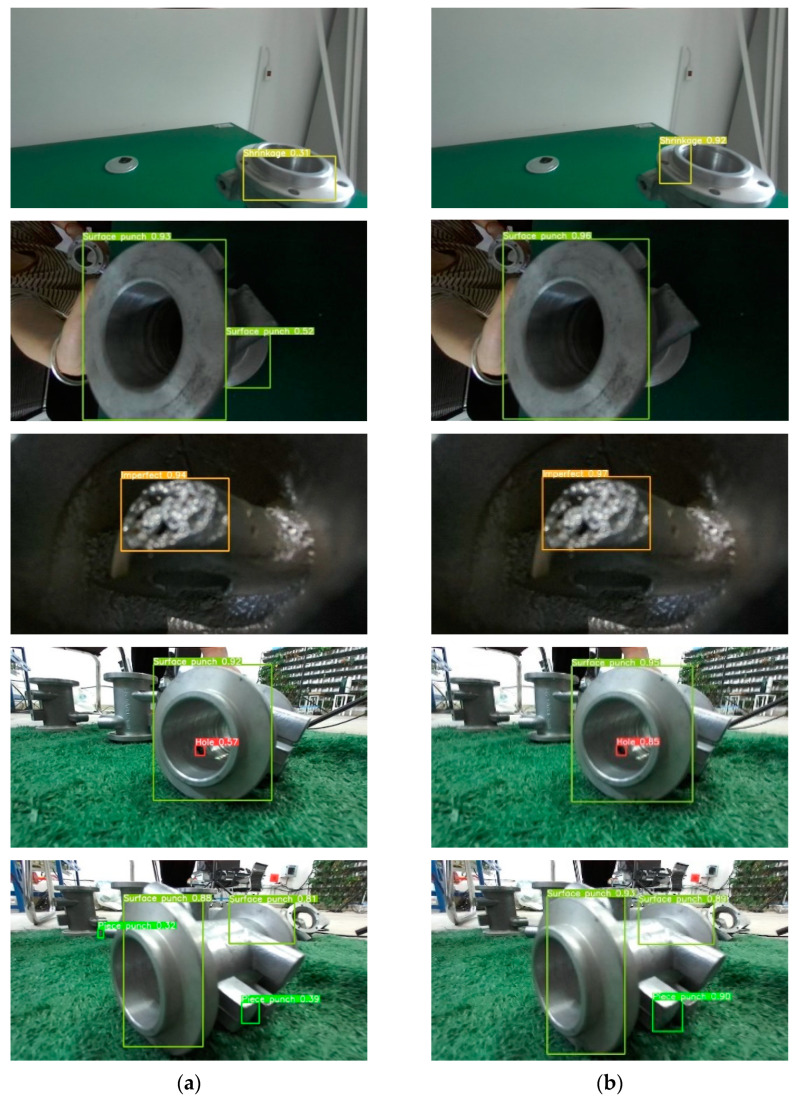
(**a**,**b**) Comparison of detection results between YOLOv8 and Improved YOLOv8 (SLGA-YOLO) on our enhanced dataset, respectively.

**Figure 11 sensors-24-04088-f011:**
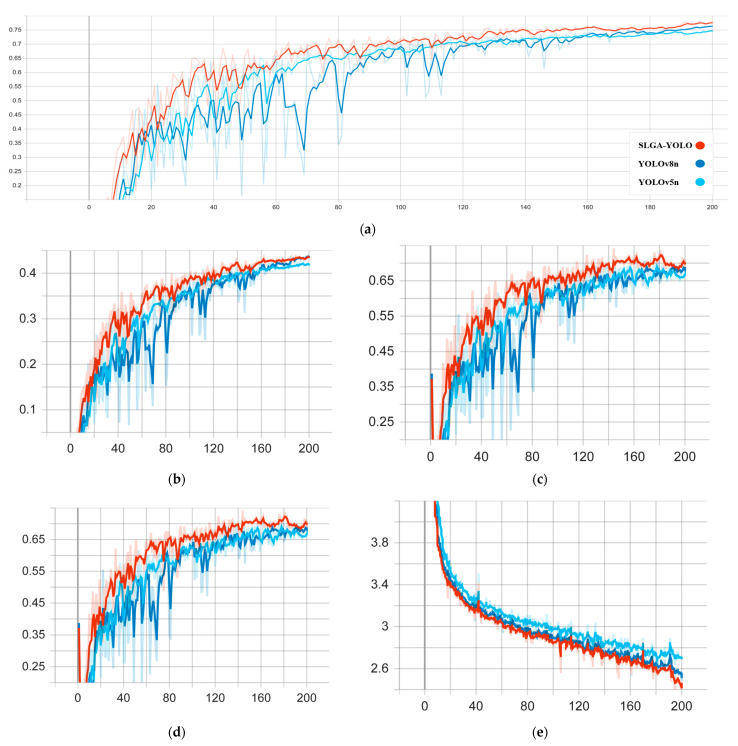
Experimental results: (**a**) mAP@0.5, (**b**) mAP@0.5: 0.95, (**c**) Precision, (**d**) Recall, and (**e**) Box_loss.

**Figure 12 sensors-24-04088-f012:**
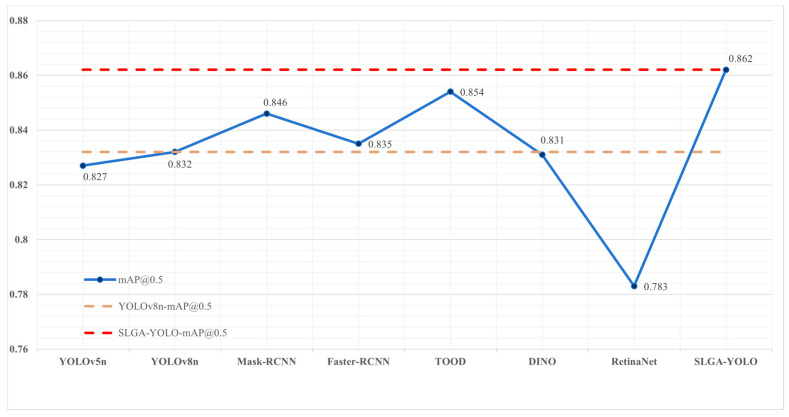
Comparative results of experiments with different models.

**Table 1 sensors-24-04088-t001:** Dataset label classification table (before enhancement).

Class	Defect Name	Datasets	Number of Labels
Hole	Sand hole	458	465
sag	Shrinkage depression	575	642
crack	Casting crack	181	182
Imperfect	Normal-casting	525	640
Shrinkage	Shrinkage	151	151
Piece punch	Piece punch	996	2671
Surface punch	Surface punch	1514	2408
Round punch	Round punch	568	670

**Table 2 sensors-24-04088-t002:** Dataset label classification table (after enhancement).

Class	Defect Name	Datasets	Number of Labels
Hole	Sand hole	514	523
sag	Shrinkage depression	1260	1333
crack	Casting crack	596	598
Imperfect	Normal-casting	527	642
Shrinkage	Shrinkage	448	448
Piece punch	Piece punch	1169	3041
Surface punch	Surface punch	1770	2774
Round punch	Round punch	594	704

**Table 3 sensors-24-04088-t003:** Model hyperparameter settings.

Parameters	Setup
Batch size	200
Image size	640 × 640
Initial learning rate	0.01
Final learning rate	0.01
Weight-decay	0.0005
Momentum	0.937

**Table 4 sensors-24-04088-t004:** Table of comparative experimental results.

Model	mAP0.5/%	FPS	Params	FLOPs/G
A	83.2	105	3,007,208	8.1
B	83.7	88	2,922,096	12.2
C	85.3	97	2,922,096	12.2
D	84.7	141	3,007,056	12.1
E	86.2	133	2,787,760	10.9

**Table 5 sensors-24-04088-t005:** Experimental results of different models on NEU dataset.

Model	mAP@0.5							FPS	FLOPs/G	Params
	All	Cr	Pa	In	PS	RS	Sc			
YOLOv5n	0.66	0.50	0.90	0.81	0.50	0.33	0.90	43.5	7.1	2504114
YOLOv8n	0.64	0.44	0.90	0.82	0.49	0.39	0.82	48.4	8.1	3007209
SLGA-YOLO	0.72	0.51	0.90	0.83	0.74	0.40	0.95	76.9	10.9	2787760

**Table 6 sensors-24-04088-t006:** Experimental results of different models on our dataset.

Model	YOLOv5n	YOLOv8n	Mask-RCNN	Faster-RCNN	TOOD	DINO	RetinaNet	SLGA-YOLO
mAP@0.5	0.827	0.832	0.846	0.835	0.854	0.831	0.783	0.862
Params/M	2.50	3.01	44.03	41.39	32.04	47.56	4.88	2.79
FLOPs/G	7.1	8.1	261	208.0	199.0	274.0	8.0	10.9

## Data Availability

The datasets generated during and/or analyzed during the current study are available from the corresponding authors upon reasonable request.

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
