# Peer review of "SLGA-YOLO: A Lightweight Castings Surface Defect Detection Method Based on Fusion-Enhanced Attention Mechanism and Self-Architecture"

_sensors, 2024, doi:10.3390/s24134088_

Round 1

Reviewer 1 Report

Comments and Suggestions for Authors

Overall, the manuscript is logically clear, easy to understand, and accurately described. However, there are two major flaws in the explanation of the methodology and the experimental comparison. Therefore, I recommend that the authors make major revisions to the manuscript.

  1. I suggest summarizing the strengths and weaknesses of existing methods after the Related Works section and explaining what motivated the authors to make these improvements to the existing YOLO framework.

  2. The authors have made several modifications to the YOLO network by adding various small modules. However, there seems to be no explanation for why these modules were added or why they were placed in specific positions. For example, why can't LSKA and SimAM switch places? The reviewer believes that simply stacking modules without justification is meaningless. The authors should clearly explain why each module is placed in its specific location and why certain modules are repeated a specific number of times. Addressing this question is crucial for evaluating the contribution of the paper.

  3. The authors have only conducted ablation experiments, and in the reviewer's opinion, testing solely on their own dataset is insufficient. Since the key innovation is algorithmic, the authors should compare their method with at least five state-of art on this topic using publicly available datasets to verify the superiority of the proposed method. (This is not a new problem, so widely used public datasets should be available).

Comments on the Quality of English Language

Good writing.

Author Response

请参阅附件

Reviewer 2 Report

Comments and Suggestions for Authors

The paper proposes a defect detection method based on an SLGA YOLO architecture for casting surface defect identification. The paper is interesting and has industrial significance. However, the justifications of the results need to be improved further.  See the comments below for further improvement of the paper.

Section 3.2

1.      The authors specify that “it is difficult to dig deep into details” for the Yolov8 model. Please clarify further in the section what these difficulties are.

2.      Please provide the reference to the LSKA attention mechanism prior to the equations.  

3.      Section 4.4. Please motivate why an improvement of 3% in the detection accuracy and 1.6 in average is significant.  

4.      Section 4.4. Often, to justify the significance, the model performance is compared to other relevant models., i.e. models specified in section 2.2. Please compare the detection accuracy to some previously applied methods.  

Comments on the Quality of English Language

Typos :

Line 30: “utilized., such as ...”

Line 45: “based 1 on ...”

Figure 3 (a) The figure is placed on top of some numbers that are not visible.

Equations 6 and 7 are not represented properly. The text and symbols overlap.

Figure 6 The plot has a poor resolution  and is not visible

Round 2

Reviewer 1 Report

Comments and Suggestions for Authors

No more comment.

Reviewer 2 Report

Comments and Suggestions for Authors

Thank you for addressing the comments. The reviewer has no further comments on the manuscript. Thank you for the interesting study.  

Comments on the Quality of English Language

The quality of the language is satisfactory.